# Deep serological profiling of the *Trypanosoma cruzi* TSSA antigen reveals different epitopes and modes of recognition by Chagas disease patients

**Guadalupe Romer**[1,2☺], **Leonel A. Bracco**[1,2☺], **Alejandro D. Ricci**[1,2☺], **Virginia Balouz**[1,2☺], **Luisa Berná**[3,4], **Juan C. Villar**[5], **Janine M. Ramsey**[6], **Melissa S. Nolan**[7], **Faustino Torrico**[8], **Norival Kesper**[9], **Jaime Altcheh**[10,11], **Carlos Robello**[3,12], **Carlos A. Buscaglia**[1,2]*, **Fernán Agüero**[1,2]*

1 Instituto de Investigaciones Biotecnológicas (IIB)–Consejo Nacional de Investigaciones Científicas y Técnicas (CONICET), B1650HMP, San Martín, Buenos Aires, Argentina, 2 Escuela de Bio y Nanotecnologías (EByN), Universidad de San Martín (UNSAM), San Martín, Buenos Aires, Argentina, 3 Laboratorio de Interacciones Hospedero-Patógeno, Unidad de Biología Molecular, Institut Pasteur de Montevideo, Montevideo, Uruguay, 4 Sección Biomatemática-Unidad de Genómica Evolutiva, Facultad de Ciencias, Universidad de la República, Montevideo, Uruguay, 5 Facultad de Ciencias de la Salud, Universidad Autónoma de Bucaramanga y Fundación Cardioinfantil—Instituto de Cardiología, Colombia, 6 Centro Regional de Investigación en Salud Pública, Instituto Nacional de Salud Pública, Tapachula, México, 7 Laboratory of Vector-borne and Zoonotic Diseases, Arnold School of Public Health, University of South Carolina, Columbia, South Carolina, United States of America, 8 Fundación CEADES, Cochabamba, Bolivia, 9 LIM-49, Hospital das Clínicas HCFMUSP, Faculdade de Medicina, Universidade de São Paulo, São Paulo, Brazil, 10 Hospital de Niños "Ricardo Gutierrez", Ciudad Autónoma de Buenos Aires, Argentina, 11 Instituto Multidisciplinario de Investigaciones en Patologías Pediátricas (IMIPP)–GCBA-CONICET, Buenos Aires, Argentina, 12 Departamento de Bioquímica, Facultad de Medicina, Universidad de la República, Montevideo, Uruguay

☺ These authors contributed equally to this work.

* cbuscaglia@iib.unsam.edu.ar (CAB); fernan@iib.unsam.edu.ar (FA)

**Data Availability Statement:** The authors confirm that all data underlying the findings are fully available without restriction. All relevant data are

## Abstract

### Background

*Trypanosoma cruzi*, the agent of Chagas disease, displays a highly structured population, with multiple strains that can be grouped into 6–7 evolutionary lineages showing variable eco-epidemiological traits and likely also distinct disease-associated features. Previous works have shown that antibody responses to 'isoforms' of the polymorphic parasite antigen TSSA enable robust and sensitive identification of the infecting strain with near lineage-level resolution. To optimize the serotyping performance of this molecule, we herein used a combination of immunosignaturing approaches based on peptide microarrays and serum samples from Chagas disease patients to establish a deep linear B-cell epitope profiling of TSSA.

### Methods/Principle findings

Our assays revealed variations in the seroprevalence of TSSA isoforms among Chagas disease populations from different settings, hence strongly supporting the differential

within the paper and its Supporting Information files.

**Funding:** Research reported in this publication was supported by the National Institute of Allergy and Infectious Diseases (NIAID) of the National Institutes of Health under award number R01AI123070 (to FA), by Agencia Nacional de Promoción de la Investigación, el Desarrollo Tecnológico y la Innovación (Agencia I+D+i, Argentina) under award numbers PICT-2009-00013, PICT-2017-0175 (to FA) and PICT-2017-3908 (to CAB), by Consejo Nacional de Ciencia y Tecnología (CONACyT, México) #261006 to JMR, and by the Brockman Medical Research Foundation (award #10008551) to MSN. The content is solely the responsibility of the authors and does not necessarily represent the official views of the National Institutes of Health. The funders did not play any role in the study design, data collection and analysis, decision to publish, or preparation of the manuscript.

**Competing interests:** The authors have declared that no competing interests exist.

distribution of parasite lineages in domestic cycles across the Americas. Alanine scanning mutagenesis and the use of peptides of different lengths allowed us to identify key residues involved in antibody pairing and the presence of three discrete B-cell linear epitopes in TSSAII, the isoform with highest seroprevalence in human infections. Comprehensive screening of parasite genomic repositories led to the discovery of 9 novel *T. cruzi* TSSA variants and one TSSA sequence from the phylogenetically related bat parasite *T. cruzi marinkellei*. Further residue permutation analyses enabled the identification of diagnostically relevant or non-relevant substitutions among TSSA natural polymorphisms. Interestingly, *T. cruzi marinkellei* TSSA displayed specific serorecognition by one chronic Chagas disease patient from Colombia, which warrant further investigations on the diagnostic impact of such atypical TSSA.

## Conclusions/Significance

Overall, our findings shed new light into TSSA evolution, epitope landscape and modes of recognition by Chagas disease patients; and have practical implications for the design and/or evaluation of *T. cruzi* serotyping strategies.

## Author summary

Chagas disease, caused by the protozoan *Trypanosoma cruzi*, is a chronic, debilitating illness of major significance in Latin America. Due to the extensive genetic and phenotypic variability of this parasite, methods able to reliably assign the infecting strain type are expected to have a positive effect on epidemiologic surveillance, prevention of transmission, and clinical management of Chagas disease. Herein we describe the diversity of human antibody specificities directed to a polymorphic *T. cruzi* antigen, TSSA, using a combination of bioinformatics and state of the art immunoassays based on peptide arrays. Our results led to the identification of novel variants of the antigen, distinct epitopes, and their key residues for antibody binding across diverse human populations and have practical implications for the design and/or evaluation of *T. cruzi* serotyping strategies.

## Introduction

Chagas disease, caused by the hemoflagellate *Trypanosoma cruzi*, is a chronic, debilitating illness of major significance in Latin America, and an emergent threat to global public health [1]. In endemic areas, *T. cruzi* shows a broad and poorly understood pattern of circulation, with partially interwoven enzootic cycles that involve hematophagous triatomine vectors and a variety of domestic and wild mammals [2,3]. Human infection primarily occurs when individuals are exposed to the excreta of infected triatomines, although other modes of transmission, such as blood transfusion, organ transplantation, ingestion of *T. cruzi*-tainted food and beverages, and congenital infection have been documented. In the last decades, different initiatives have been undertaken in most endemic countries, with a concomitant decrease in the actual numbers of acute infections [1]. However, in the absence of vaccines or appropriate drugs for large-scale intervention, improved diagnostic methods are urgently needed for the effective implementation of disease control programs.

Chagas disease may range from subclinical to severe and often fatal cardiac and/or intestinal complications [1]. Such variable clinical presentation has been attributed in part to the extensive genetic polymorphism from both the exposed human population and the infecting parasite. Indeed, *T. cruzi* is currently understood to comprise six genetic lineages or discrete typing units (DTU), termed TcI to TcVI [2,3]. A potential seventh DTU, TcBat, has been recently identified in South and Central American bats [4]. The evolutionary relationships among DTUs have not been fully established, but it is accepted that TcI, TcII, TcIII and TcIV have more ancient origins whereas TcV and TcVI are clusters of hybrid strains, the product of relatively recent genetic crosses between TcII and TcIII parentals [2,3]. *T. cruzi* DTUs display distinct geographical distribution and maintain ecological associations with different spectra of competent triatomines and/or susceptible mammals [2,5]. Though still controversial, parasite genetic variability may also correlate with relevant phenotypic traits, including vector infectivity, vector-to-mammal transmissibility, susceptibility to drugs, and disease outcome [6–9]. In this context, methods able to reliably assign the infecting strain type directly from biological samples are expected to have a positive effect on epidemiologic surveillance, prevention, and clinical management of Chagas disease.

The trypomastigote small surface antigen (TSSA) is a parasite surface adhesin shown to elicit robust antibody responses during *T. cruzi* infections [10–12]. Sequencing of TSSA alleles from different strains revealed several diagnostic polymorphisms that allowed their classification into four 'variants', termed TSSAI, TSSAII, TSSAIII and TSSAIV, each one corresponding to an ancestral DTU [13]. In agreement with their evolutionary history, strains from hybrid DTUs (TcV and TcVI) code for both TSSAII and TSSAIII alleles [13,14]. Most TSSA sequence variations map to the protein mature region, i.e., the sequence displayed on the parasite surface upon processing of the secretory signals, and affect TSSA adhesive properties [10,15,16]. Interestingly, such polymorphisms were also shown to alter TSSA antigenic structure, which led to the proposal that antibody signatures to this molecule may be exploited for indirect, serological typification of *T. cruzi* infections [10].

Surveys exploring the discrimination power of TSSA antibody responses led to numerous insights into parasite eco-epidemiology and showed that this molecule has the potential to predict the infecting parasite strain type with near DTU-level resolution [12,17–20]. In human infections, TSSAII is the variant with the highest seroprevalence [10,20,21], with an estimated ~80–95% reactivity in Southern Cone countries (Argentina, Bolivia, Brazil, Paraguay, Chile) where TcII/TcV/TcVI strains prevail [22–25]. TSSAIII antibodies, on the other hand, were rarely identified in Chagas disease patients, in line with genotypic estimates showing the main sylvatic circulation of TcIII strains [2,3]. The same holds true for TSSAIV antibodies, despite TcIV DTU being the second most common cause of Chagas disease in Venezuela and Colombia. TSSAI also showed a consistently low seroprevalence among *T. cruzi*-infected human populations [10,20,21]. This is at odds with TcI broad distribution and fairly frequent occurrence in human infections and may be attributed to TSSAI poor intrinsic immunogenicity or to protein glycosylation issues [26].

Previous mapping efforts highlighted a ~40 amino acids-long immunoreactive region spanning most of the TSSAII mature sequence [27,28]. The length of this region, which largely exceeds the size range determined for canonical linear B-cell epitopes (5–7 amino acids) [29], suggested a complex antigenic structure for this molecule. Building on this knowledge and aiming to improve the serotyping performance of TSSA, we herein used a combination of immunosignaturing approaches based on peptide microarrays and serum samples of Chagas disease patients from different areas to establish an exhaustive linear B-cell epitope fingerprinting of TSSA.

## Methods

### Ethics statement

The approval of the different institutional review boards as well as the informed consent of the individuals that provided the samples analyzed in this study can be found in [30]. The Institutional Review Board of UNSAM has evaluated the current project and considered that it complies with the Basic HHS Policy for Protection of Human Research Subjects requirements to be included in the 'exemption 4', because it involved the use of de-coded and de-identified human serum samples obtained from sera repositories where they were preserved for diagnosis studies purposes (Institutional Review Board, Foundation for Research in Biotechnology, Approved Protocol 001/2016, Project: High-throughput epitope discovery: use of next-generation peptide chips for fast identification and fine mapping of diagnostic and prognostic markers for Chagas disease).

### Study population

Serum samples evaluated in this work were from subjects coursing the chronic phase of Chagas disease, with no cardiac involvement or other disease-associated pathology ($n = 71$), or from healthy individuals from the same geographic region ($n = 45$). These subjects originated from 6 different countries across the Americas. All subjects were assayed individually, and some of them were also assayed as part of pooled serum samples (one pool per country). Each sample was given a new name such as AR_P1 or BO_E2, where the first two letters indicated their country of origin ('AR' = Argentina; 'BO' = Bolivia; 'BR' = Brazil; 'CO' = Colombia; 'MX' = Mexico; 'US' = United States) and the 'P' in the suffix indicated that it was one of the sera used to generate the different pools (otherwise the suffix would be an 'E'). Written informed consent was obtained for each subject, and samples were decoded and de-identified before they were provided for research purposes. The complete information of the population study can be found in S1 Table and in [30].

### Microarray design

Two microarray designs were used in this work. The first design contained three sets of peptides: i) 15mers spanning the complete sequences of *T. cruzi* TSSA reference variants (as defined in [13], Fig 1); ii) 15mers spanning residues 30 to 55 of TSSAII and bearing 1 to 10 substitutions as compared to the reference sequence; and iii) peptides of different length (13mers to 8mers) encompassing residues 24 to 62 of TSSAII. This first design was assayed with the 6 pooled sera, one per country. The second design contained i) 16mers spanning all four TSSA reference variants; and ii) mutated 16mers spanning residues 31 to 50 of TSSAII, in which each position was replaced by Alanine (or Glycine, in the case of a naturally occurring Alanine). This second design was assayed with 71 individual serum samples. Arrays were designed as consecutive sets of k-length peptide subsequences (kmers) with an offset of 1 residue and an overlap of length = k-1, therefore ensuring maximal resolution. Because of this extensive overlapping, consecutive peptides on each set act as internal pseudo-replicas [27,30]. Peptides were named by the identity and relative number of the residue occupying its middle position (taking as 1 the Methionine proposed as site of TSSA translation initiation [10]) (Fig 1). Further details of peptide microarrays used in this work can be found in the S2 Table and at the ArrayExpress database Functional Genomics Data Collection (https://www.ebi.ac.uk/biostudies/arrayexpress/arrays/) under accession codes A-MTAB-692 and A-MTAB-693 (Array Designs); and E-MTAB-11651 and E-MTAB-11655 (Assay Data).

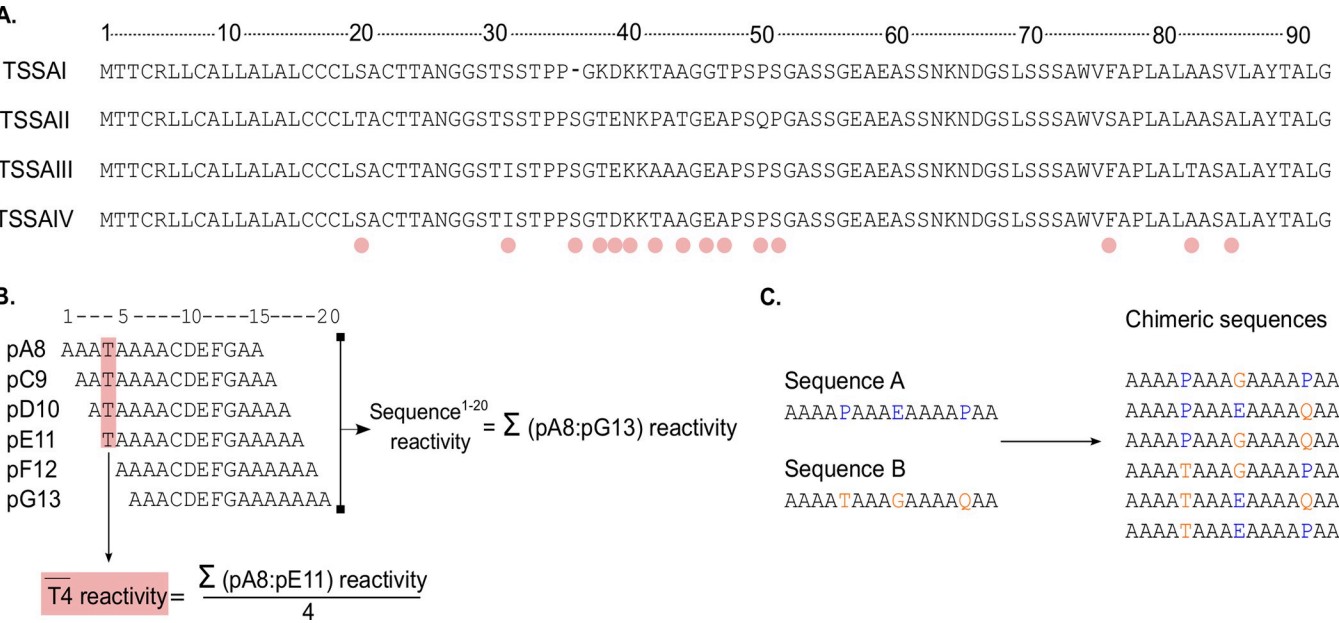

**Fig 1. *Microarrays design and reactivity evaluation*. A)** Complete sequence of *T. cruzi* TSSA reference variants (TSSAI [AFF60282.1], TSSAII [XP_808931.1], TSSAIII [TXP_805410.1] and TSSAIV [AAQ73324.1]). Polymorphic positions are indicated with pink dots. **B)** Diagram showing peptide nomenclature, and the calculations of the overall reactivity of a specific sequence and residue. Peptides were named by the identity and relative number of the residue occupying its middle position (taking as 1 the Methionine proposed as site of TSSA translation initiation), using standard single-letter abbreviations. **C)** Diagram showing chimeras generated from variable sequences.

## Microarray synthesis, screening, data analysis and presentation

The overall synthesis, screening and downstream processing procedures of high-density peptide microarrays have been extensively described [27,30–32]. Briefly, two replicas of each array were probed with serum samples from 71 individuals infected with *T. cruzi* or 6 pools of sera prepared from some of these samples. Readouts for pooled or individual samples were normalized independently using quantile normalization [30]. Data analysis and presentation were done with *ad hoc* R scripts [33] and ggplot2 package [34]. The sequences and raw fluorescence values for the peptides that were evaluated in this study and the R scripts can be found at S1 to S3 Files and at https://github.com/trypanosomatics/The-Chagas-Disease-Antigen-and-Epitope-Atlas/tree/main/77_romer_et_al_TSSA. Reactivity for each peptide was calculated as its mean fluorescence value recorded in duplicated arrays. The overall reactivity of a specific sequence was calculated as the sum of reactivities of its constituent peptides; and the reactivity for each residue in a specific sequence was calculated as the mean signal for all peptides containing such residue (Fig 1). It is worth noting that this parameter is calculated upon a different number of signals depending on the relative position of the evaluated residue. In mutational scanning assays, the net impact caused by a specific replacement was calculated as the difference in reactivity recorded for the original and the surrogate residue.

## In silico predictions and phylogeny analyses

Genomes of *T. cruzi* isolates B. M. Lopez (TcI), Brazil A4 (TcI), Colombiana (TcI), Dm28c (TcI), G (TcI), JR cl. 4 (TcI), Sylvio X10/1 (TcI), Berenice (TcII), 11 (TcII), S15 (TcII), S23b (TcII), S44a (TcII), S92a (TcII), Esmeraldo (TcII), Ikiakarora (TcIII), 231 (TcIII), Bug2148 (TcV), SC43 (TcV), CL (TcVI), CL Brener (TcVI), TCC (TcVI) and Tula cl2 (TcVI), as well as *T. cruzi marinkellei*, *T. brucei*, *T. congolense*, *T. evansi*, *T. grayi*, *T. rangeli*, *T. theileri*, *T. vivax*,

*Blechomonas ayalai*, *Bodo saltans*, *Crithidia fasciculata*, *Endotrypanum monterogeii*, *Leishmania spp*, *Leptomonas pyrrhocoris*, *L. seymouri* and *Paratrypanosoma confusum* were retrieved from NCBI (https://www.ncbi.nlm.nih.gov/genome/browse#!/eukaryotes/25/) and TritrypDB (http://tritrypdb.org/tritrypdb). *T. cruzi* RA (TcVI) genome was recently obtained in our lab using PacBio RSII technology. *T. cruzi* Y (TcII), MT3663 (TcIII), Jose Julio (TcIV) and BOL-FC10A (TcV) genomes were obtained at the Instituto Pasteur, using PacBio and Nanopore technologies. All these genomes were mined using BLASTn and the nucleotide sequence of CL Brener TSSAII (TcCLB.507511.91) as bait. Given the conservation between TSSA and TcMUC genes on its deduced signal peptides (SP) and glycosyl phosphatidylinositol (GPI)-anchoring signals [35], results had to be filtered from spurious hits using two parameters: the length (276 to 279 bp) and identity (>85%) of the alignments. Retrieved sequences were translated with transeq (EMBOSS) [36]. After manual curation of the output, i.e., removal of pseudogenes and/or truncated sequences and filtering for redundancy [37], phylogenetic trees were built using ClustalW [36] and IQ-TREE [38]. TSSA reference variants (TSSAI [AFF60282.1], TSSAII [XP_808931.1], TSSAIII [TXP_805410.1] and TSSAIV [AAQ73324.1]), were included to define clusters of sequences. The final phylogram is the consensus tree of 1,000 bootstrap replicates and was graphically modified for presentation using iTOL [39]. Identification of SP and GPI-anchoring signals was done using the online servers SignalP 4.0 and PredGPI, respectively. Further details of TSSA sequences used in this work can be found in the S3 and S4 Tables.

## Recombinant TSSA-based ELISA (TSSA-ELISA)

The glutathione S-transferase (GST)-fusion proteins bearing the central region (residues 24 to 61–62) of TSSAI, TSSAIII and TSSAIV were obtained by PCR from genomic DNA of different strains of *T. cruzi* [31] whereas the sequence corresponding to residues 30–50 from *T. cruzi marinkellei* TSSA was custom-synthesized (Macrogen). Further details of these GST-fusion proteins can be found in the S5 Table. GST and a GST-fusion protein bearing a fragment of the highly immunoreactive *T. cruzi* antigen 1 (GST-Ag1, [40,41]) were used as controls. These proteins were expressed in *E. coli* and purified from the soluble fraction to almost homogeneity by a single glutathione affinity chromatography step as described [42]. Flat-bottomed 96-well Nunc-Immuno plates (Nunc, Roskilde, Denmark) were coated overnight at 4˚C with 80 μl of each GST-fusion protein dissolved in carbonate buffer (pH 9.6) at 0.25 μg/ml and processed for a previously validated, colorimetric TSSA-ELISA [42]. Serum samples were assayed in triplicate at 1:100 dilution.

## Results

### Differential antibody recognition of TSSA variants across the Americas

We designed 4 sets of consecutive 16mer peptides with 1-residue offset and 15-residue overlaps that, together, encompassed the complete sequence of TSSA reference variants (Fig 1). This array was probed with sera of chronic Chagas disease patients from different settings, hence likely infected by distinct *T. cruzi* strains/DTUs [2,3,5]. Peptide fluorescence signals were recorded and used to calculate the reactivity for each serum and each TSSA variant (Fig 2A). Overall, TSSA displayed ~69% seroprevalence (49 positive out of 71 tested samples) among the evaluated population, though this figure increased up to 97.2% (35 positive out of 36) when only samples from Southern cone countries (Argentina, Bolivia, and Brazil) were considered (Fig 2B). The primary reactivity for these latter samples was almost invariably against TSSAII, with some of them displaying minor, secondary reactivity towards TSSAIII and/or TSSAIV variants (Fig 2B). Consistent with previous data [10,20], only a few Southern

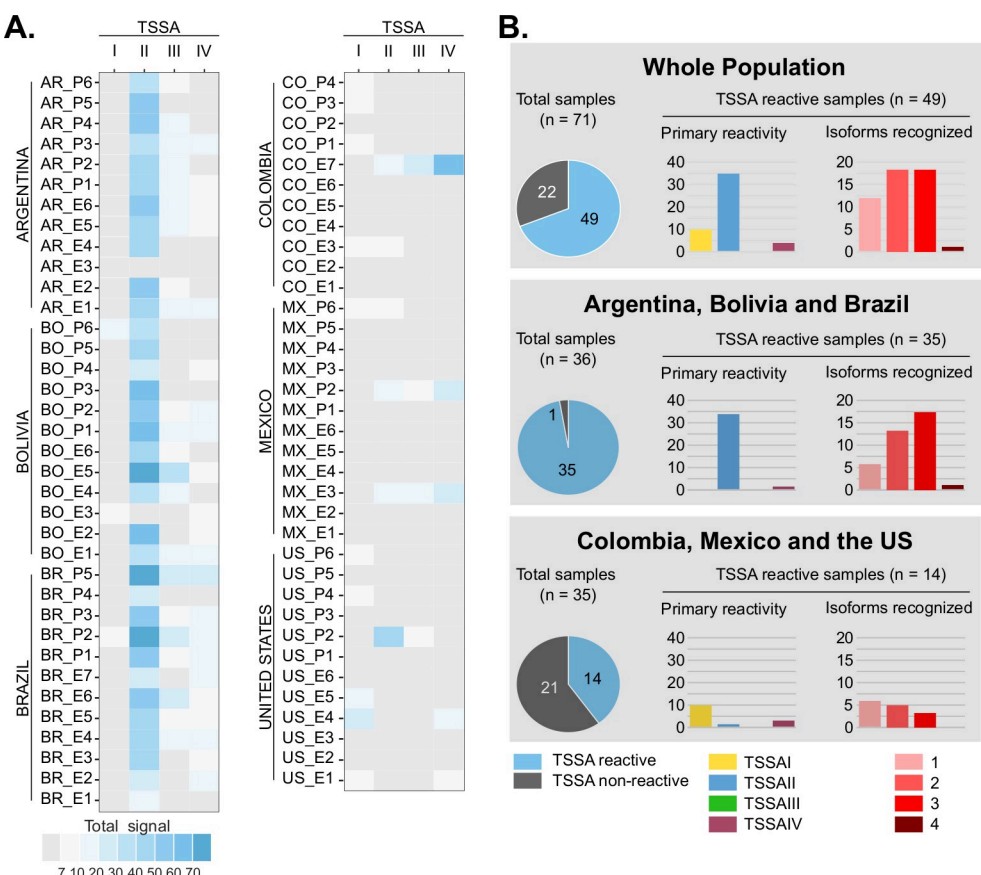

**Fig 2.** *Differential seroprevalence of TSSA variants across the Americas.* **A)** Arrays made up of consecutive 16mer peptides and spanning the complete sequences of TSSA reference variants were probed with serum samples from chronic Chagas disease patients of different geographic origin, and the reactivity of each TSSA (indicated with a color scale) was calculated as the sum of the reactivities of its constituent peptides. **B)** TSSA reactivity features of the population analyzed as a whole (top), or upon stratification in Southern cone countries' samples (middle) and Northern South America and North American samples (bottom). In every case, the overall TSSA seroprevalence is indicated by pie charts to the left. The bar charts show the primary reactivity (the most reactive TSSA isoform, left chart), and the number of isoforms recognized by TSSA-reactive sera (right chart). The *y*-axis refers to the number of samples.

cone countries' samples (BO_P6, BO_E3, BR_P2) reacted (and very faintly) with TSSAI (Fig 2A).

Serological evaluation of samples from Colombia, Mexico, and the US revealed a completely different scenario. TSSA was seldom recognized by these sera, with only 14 out of 35 analyzed samples (40%) showing reactivity (Fig 2). Such reactivities were weak as compared to samples from Southern cone countries and, most importantly, presented different specificities. Serum samples showing robust anti-TSSA signals (CO_E7, MX_E3, MX_P2 and US_P2) also displayed mixed recognition of TSSAII, TSSAIII and/or TSSAIV molecules. However, and in contrast with sera from Argentina, Brazil, and Bolivia that showed primary reactivity towards TSSAII, 3 of these samples (CO_E7, MX_E3, MX_P2) yielded maximal signals for TSSAIV (Fig 2B). The remaining TSSA-positive samples from Colombia, Mexico and the US displayed primary though barely detectable reactivity to TSSAI (CO_E3, CO_P1, CO_P3, CO_P4, MX_P6, US_E1, US_E4, US_E5, US_P4, US_P6), with some of them also showing secondary reactivity towards TSSAII (CO_E3, MX_P6) or TSSAIV (US_E1, US_E4) (Fig 2).

## Linear B-cell epitope fingerprinting in TSSAII

For each serum and TSSA variant we also calculated the reactivity of individual residues and used these values to reconstruct the immunoreactivity profiles of the entire molecules (S1 Fig). These analyses revealed an extensive inter-individual variability in the TSSAII antibody signatures, thereby suggesting a complex antigenic structure for this molecule. To get further insights into this issue, sets of different kmers (15mers, 13mers, 12mers, 11mers, 10mers, 9mers, and 8mers) encompassing residues 24 to 62 of TSSAII (TSSAII[24-62]) were serologically evaluated. Instead of using individual samples as before, these arrays were probed with pools of sera from different settings (S2 Table). As expected from individual reactivities (Fig 2), pools of sera from Mexico and Colombia yielded background signals against TSSAII[24-62] and were thus excluded from further analyses. On the contrary, pools of sera from Argentina, Bolivia, Brazil, and the US (most likely driven by the TSSAII-reactive sample US_P2 included in this latter pool, Fig 2) displayed strong reactivity against TSSAII[24-62] (Fig 3). Positive signals for all of them started at peptides pS32-pP35, depending on the kmer length. Peptides located 1 residue to the right from those positions displayed a remarkable increase in their signals and those located 2 residues to the right reached maximal or near-maximal values (~5 arbitrary units of fluorescence). For larger kmers, and notwithstanding the geographic origin of the pool, a continuous high reactivity plateau extending from peptides pP34 or pP35 and up to peptides pP48-pQ50 could be outlined (Fig 3). Peptides located *C*-terminally to pP48-pQ50 (up to peptides pP51-pA53 for Argentina, Bolivia, and the US pools or to pA58 for the Brazilian pool) showed a steady decline in their reactivity, hence marking the end of the immunoreactive region. Signals recorded for shorter peptides, however, revealed that the high reactivity plateau may be decomposed into two 'peaks' placed at its *N*- and *C*-terminal ends, which flanked a 'valley' centered around peptide pK41 (Fig 3). This pattern, particularly evident for the Bolivia, Brazil, and the US pools, is consistent with the presence of two major and discrete B-cell linear epitopes.

Supporting the *N*-terminal antigenic peak (henceforth epitope A), Bolivia and the US pools highlighted the sequences 36-SGTEN-40 and 37-GTEN-40, respectively, whereas both Argentina and Brazil pools highlighted the sequence 35-PSGTEN-40. As for the *C*-terminal antigenic peak (henceforth epitope B), Brazil and the US pools revealed the sequences 42-PATGEAP-48 and 42-PATGEA-47, respectively, whereas Argentina and Bolivia pools highlighted the sequence 42-PATGEAPS-49 (Fig 3). Overall, we tentatively defined 35-PSGTEN-40 and 42-PATGEAPS-49 sequences as epitopes A and B, respectively (Fig 3). Solely the Brazilian pool, and most likely due to the inclusion of the broadly reactive serum BR_P2 (S1 Fig), revealed a weak, third peak (defined as epitope C) encompassing the sequence 53-ASS-55 (Fig 3).

## TSSA natural variability

Previous efforts to estimate TSSA diversity were limited to sequencing a few alleles from selected *T. cruzi* strains [10,13]. To reveal the full extent of TSSA natural variability, we carried out comprehensive searches of trypanosomatids' genomic databases. These screenings yielded a total of 45 *T. cruzi* genomic *loci* containing sequence/s showing significant similarity (>90% nucleotide identity) to the *T. cruzi* CL Brener TSSAII (TcCLB.507511.91) used as bait (S3 Table). Following exclusion of redundant and/or truncated sequences and pseudogenes, we identified 9 novel TSSA sequences, which were classified as variants of TSSAII, TSSAIII, TSSAIV or TSSAI by similarity clustering with reference sequences (Fig 4A). One TSSA from the Jose Julio (TcIV) strain was tentatively assigned as a TSSAIV variant although it did not cluster robustly with the TSSAIV reference sequence (Fig 4A). Our screening also retrieved one TSSA sequence (85.16% nucleotide identity to TcCLB.507511.91) from the bat-restricted trypanosome *T. cruzi marinkellei* (termed TcMARK TSSA). This was the most divergent TSSA

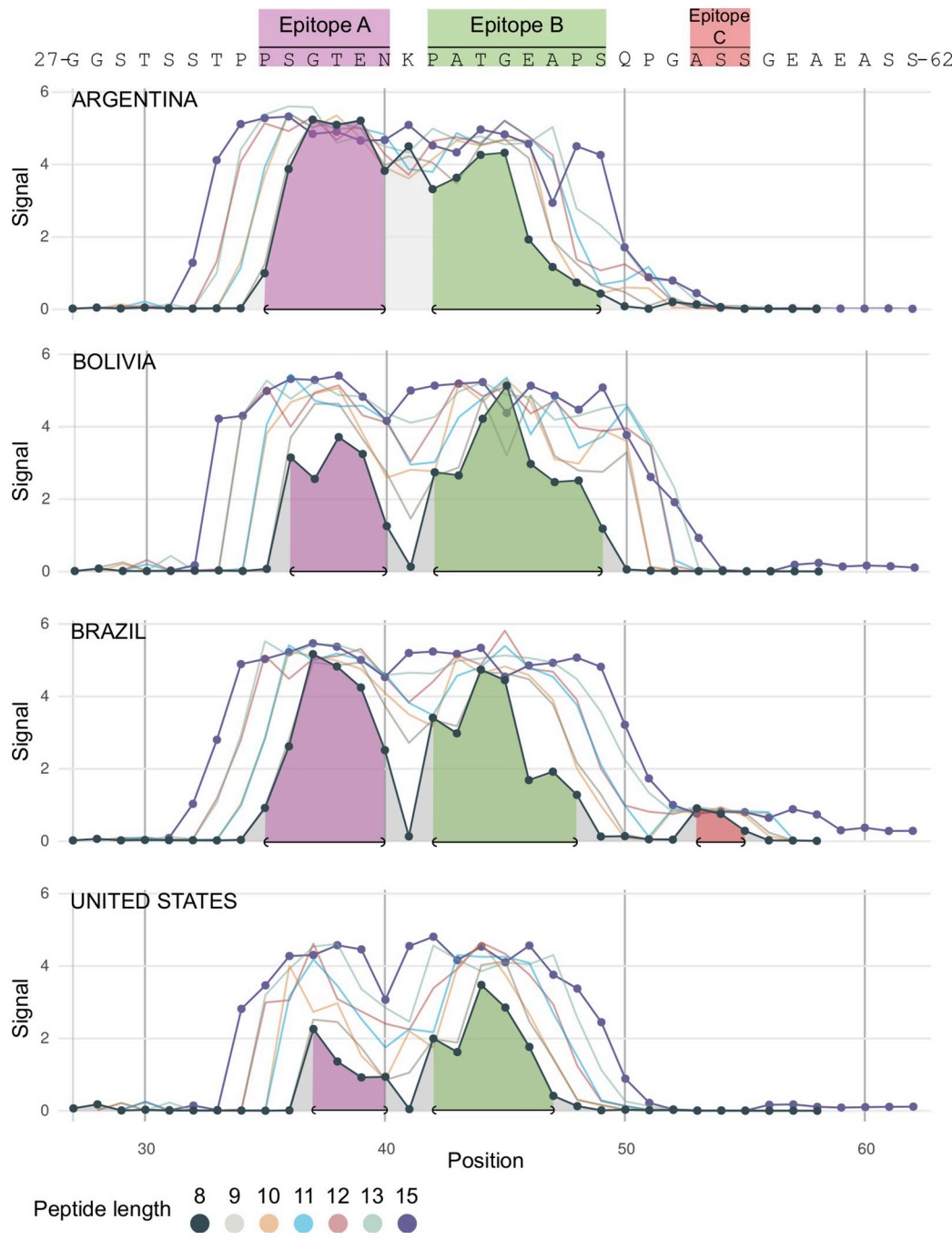

**Fig 3. B-cell epitope fingerprinting in TSSAII.** Peptide arrays comprising distinct sets of consecutive kmers, overlapped by k-1 residues and spanning the sequence of TSSAII[24-62] (or the complete TSSAII sequence for 15mers) were probed with the indicated pools of *T. cruzi*-infected sera. The mean reactivity of each peptide is shown. Residues included in epitopes A and B, as defined by 8mers arrays, are indicated in purple and green, respectively. Epitope C (in pink) was solely highlighted by the Brazilian pool of sera. Reactivities for peptides pE59 to pS62 were calculated solely with 15mer data.

and defined a separated tree branch (Fig 4A). Notably, and despite certain discordant cases, i.e., 1 TSSAI in one clone of Y (TcII), 2 TSSAI sequences in Bug2148 (TcV) and 1 TSSAII in Ikiakarora (TcIII) there was a strict correlation between the identified TSSA isoform(s) and the reported DTU of the strain in which they were found (Fig 4B, S3 and S4 Tables).

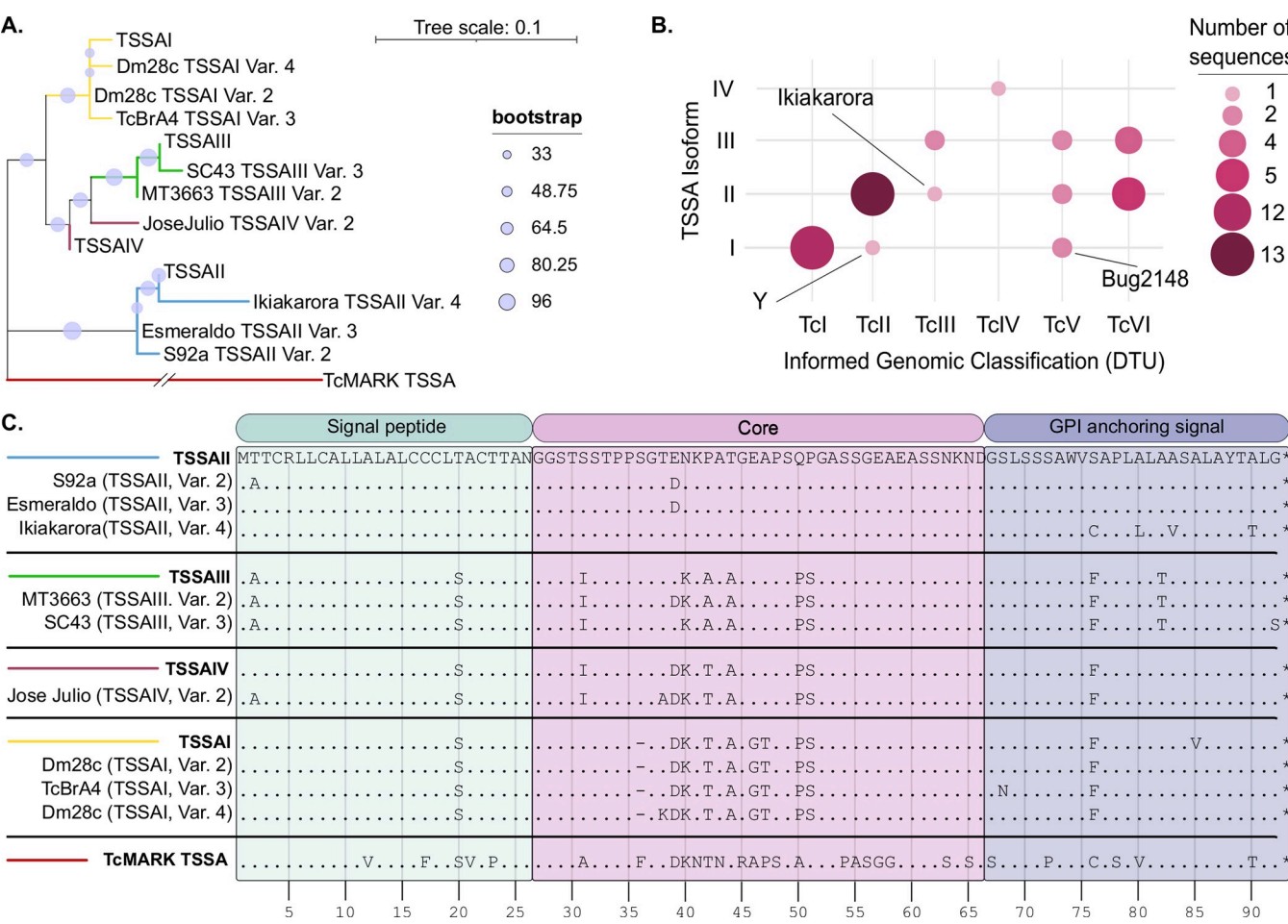

**Fig 4. Phylogenetic relationships and features of TSSA proteins. A)** Unrooted phylogenetic tree constructed from deduced TSSA amino acid sequences using the Maximum Likelihood estimation based on p-distance (TcMARK TSSA distance = 0.434). Bootstrap values are represented by a light blue dot on each branch. **B)** Correlation plot showing, for each unique TSSA sequence, the isoform assigned and the informed genomic classification of the strain in which it was found. The number of sequences per isoform/DTU combination is indicated with size-scaled dots. **C)** Comparison of the deduced amino acid sequences from TSSA variants identified in this work and TSSA reference sequences. Amino acid identity with the TSSAII reference sequence is indicated by dots. The predicted signal peptide and GPI anchor are indicated.

Protein alignments revealed a total of 20 polymorphic positions across *T. cruzi* TSSAs, 11 of which occur in their central and antigenic region (Fig 4C). As compared to the TSSAII reference sequence, these polymorphisms include positions 31 (S-to-I in TSSAIII and TSSAIV variants), 38 (T-to-A or T-to-K in certain TSSAIV and TSSAI variants, respectively), 39 (E-to-D in TSSAIV, TSSAI and certain TSSAII and TSSAIII variants), 40 (N-to-K in TSSAIII, TSSAIV and TSSAI variants), 42 (P-to-A in TSSAIII variants or P-to-T in TSSAIV and TSSAI variants), 44 (T-to-A in TSSAIII, TSSAIV and TSSAI variants), 46 (E-to-G in TSSAI variants), 47 (A-to-T in TSSAI variants), 50 (Q-to-P in TSSAIII, TSSAIV and TSSAI variants) and 51 (P-to-S in TSSAIII, TSSAIV and TSSAI variants). All TSSAI variants, in addition, bore a deletion of the otherwise conserved Ser36 (Fig 4C).

## TSSA diversity and antibody recognition

To assess the impact of TSSA diversity on antibody recognition, tiling sets of consecutive 15mer peptides spanning residues 30 to 55 of TSSAII (TSSAII$^{30-55}$) were synthesized. Each set

of peptides bore either single or combined (2 to 10) substitutions at polymorphic positions 31, 38 (solely the T-to-K permutation), 39, 40, 42, 44, 46, 47, 50 and 51. To simplify the array's layout, all of them were designed with a Ser residue at position 36. Upon serologic evaluation of these arrays with above-described pools, the overall reactivity of chimeric sequences was calculated as before. Among single-substituted sequences, those bearing E46G, P42T or T38K displayed reduced reactivity with the TSSAII-reactive pools (Argentina, Bolivia, Brazil, and the US) (Fig 5A). Other substitutions, such as P42A, T44A, and A47T led to reduced reactivity when probed with the Bolivia, Argentina, and the US pools but not with the Brazilian pool. In the case of P42, its replacement by Threonine (P42T) had a more pronounced effect on reactivity than its replacement by Alanine (P42A). N40K substitution, on the other hand, displayed reduced reactivity as compared to the TSSAII reference sequence solely for the Bolivia and the US pools (Fig 5A). Interestingly, this substitution, which is distinctive of TSSAIII, TSSAIV and TSSAI variants (Fig 4C), led to a notable increase in the signal of the Mexico pool (Fig 5A). Though not proven, this effect was likely driven by the TSSAIV-reactive MX_P2 serum included in such a pool (Fig 2). Overall, it could be concluded that T38K, P42A/T, T44A, E46G, and A47T impose major constraints on TSSAII antibody recognition by most Chagas disease sera and may thus be classified as 'relevant' substitutions. The remaining substitutions (S31I, E39D, Q50P, and P51S) did not affect TSSAII reactivity and may be thus classified as 'non-relevant'. N40K could be classified either as relevant or non-relevant depending upon the geographic origin of the evaluated pool.

For each single-substituted sequence we also calculated the reactivity of individual residues and used these values to reconstruct the immunoreactivity profiles of the entire molecules. From these analyses, it emerged that relevant substitutions do not affect the recognition of the entire TSSAII[30-55] sequence but, rather, either of its *N*-terminus (T38K, N40K [solely for the Bolivia and the US pools]) or its *C*-terminus (P42A/T, T44A, E46G and A47T) (Fig 5B). Interestingly, these data strictly correlated with our previous epitope proposal: T38 and N40 were included in the *N*-terminally located epitope A (35-PSG**TEN**-40) whereas P42, T44, E46 and A47 were all included in the *C*-terminally located epitope B (42-**P**A**T**G**EA**PS-49).

As previously verified when assayed individually (Fig 5A), different combinations of non-relevant substitutions had a negligible effect on the overall TSSAII[30-55] reactivity, suggesting that their impact was neither additive nor cooperative (Fig 5C). Conversely, accumulation of relevant substitutions had a differential effect on sequence reactivity depending on the specific combinations. Sequences bearing 2, 3 or up to 4 substitutions included in epitope B (P42A/T, T44A, E46G and A47T) showed a rather similar reactivity than those single-substituted (Fig 5C). In all cases, and as expected from their individual evaluation (Fig 5B), only the *C*-terminal reactivity of sequences bearing such combinations was affected (S2 Fig). Solely upon combination of any (or a group) of epitope B-located substitutions with an epitope A-located substitution, i.e., T38K, reactivity of the ensuing sequence was completely abolished (Figs 5C and S2).

We finally evaluated TSSAII-reactive pools against all chimeric sequences, bearing every possible combination of substitutions (Fig 5D). Puzzlingly, these assays revealed that E39D, initially classified as non-relevant, and N40K, initially classified as non-relevant for Argentina and Brazil pools (Fig 5A), do have an inhibitory effect on TSSAII[30-55] reactivity in the context of sequences in which epitope B was affected, i.e., sequences showing substitutions (or a combination of substitutions) at positions P42, T44, E46 and/or A47. The inhibitory effect of E39D and N40K replacements became more evident in sequences bearing 3 or 4 substitutions in such positions, where 3 groups of sequences could be outlined based on their mean reactivity (Fig 5D). The most reactive sequences corresponded to those bearing a 'TSSAII configuration' at the tripeptide 38-TEN-40. The second group, showing intermediate reactivities, presented a Threonine at position 38 combined with either E39D or N40K substitution whereas the third

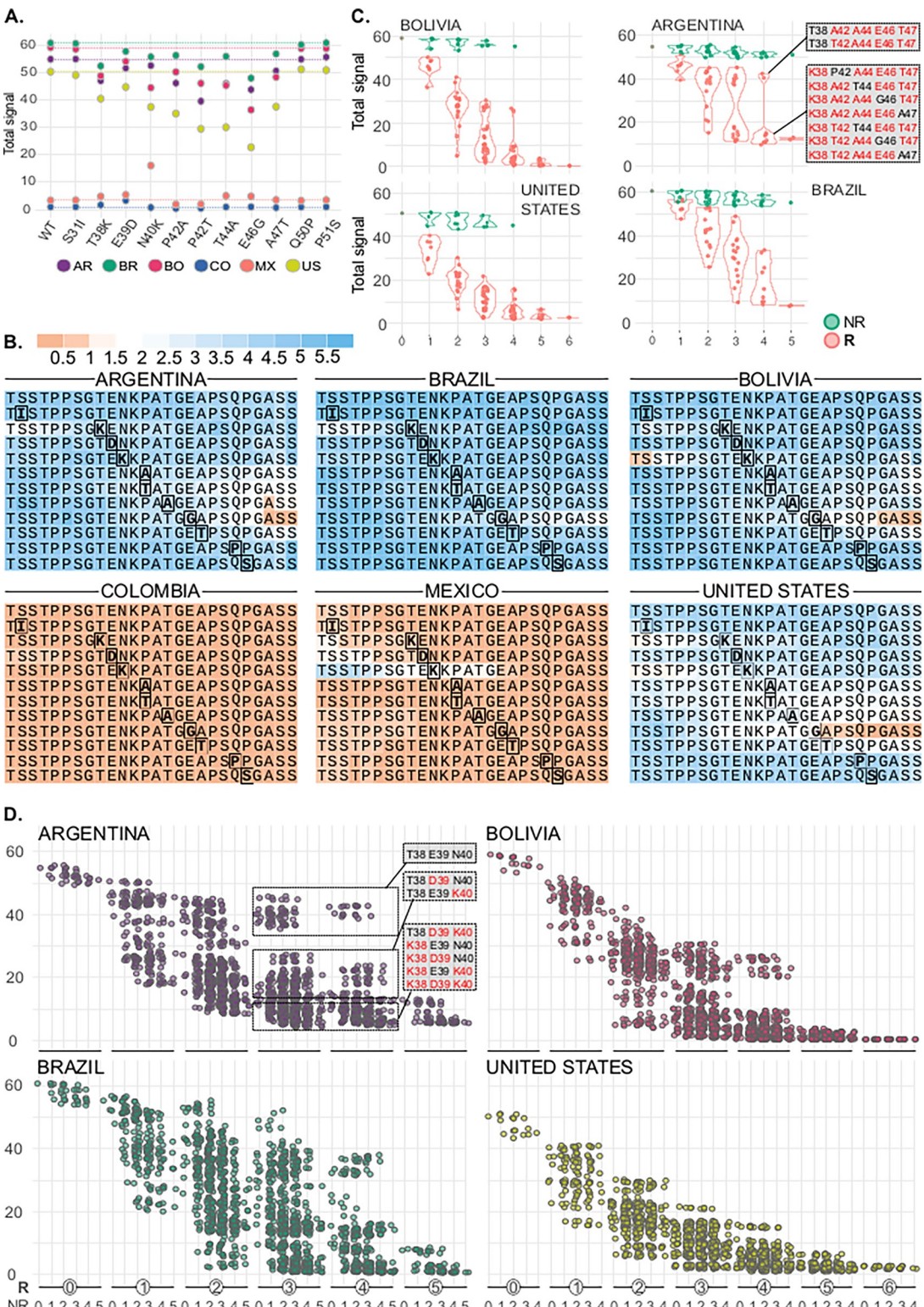

**Fig 5. *TSSA polymorphisms and antibody recognition.*** Peptide arrays comprising sets of fully overlapped 15mers, spanning residues 30 to 55 of TSSAII reference variant (TSSAII[30-55]) and bearing substitution(s) at polymorphic positions were probed with pools of chronic Chagas disease sera of different geographic origin. **A)** Reactivity of wild-type (wt) and single-substituted sequences for each pool. **B)** Mean reactivity of each residue in the context of wt (top row) or single-substituted sequences is indicated with a color scale. For each sequence, the replaced residue is marked in bold. **C)** Reactivity of sequences bearing

combinations of either non-relevant (NR, green dots) or relevant (**R**, red dots) substitutions. Amino acid identity at the indicated positions in sequences bearing 4 relevant substitutions (for the Argentina pool sample) is shown in the boxes, where original and replaced residues are depicted in black and red, respectively. **D**) Reactivity of sequences bearing every possible combination of **R** and NR substitutions. For the Argentina pool sample, the identity of residues at positions 38, 39 and 40 from molecules displaying high (upper box), intermediate (middle box) or low reactivity (lower box) is indicated as in (C).

and less reactive group of sequences presented both E39D and N40K substitutions (Fig 5D). Reactivity of these latter sequences was barely detectable, thus in the range of those bearing a T38K substitution (Fig 5D).

## TSSAII mutational scanning

To assess the contribution of each amino acid to TSSAII antibody recognition, we performed a complete Alanine mutational scan of residues 31 to 50 of this molecule. These arrays were independently evaluated with 32 sera showing primary reactivity towards TSSAII (Fig 2) and the impact of a specific replacement calculated as the difference between the mean reactivity of the original and the Alanine-substituted (or Glycine-substituted, in case of A43 and A47 positions) sequences (Fig 6A). The overall serum signature derived from these assays indicated that most of the replacements (P34A, P35A, S36A, G37A, T38A, E39A, N40A, K41A, P42A, A43G, T44A, G45A, A47G, P48A and Q50A) diminished TSSAII reactivity (Fig 6B). Among them, G37A, T38A, E39A, N40A, K41A, P42A and P48A yielded a more dramatic effect, strongly suggesting that these positions play a critical role in antibody pairing (Fig 6B). We also calculated the difference in the mean reactivity at each position between the natural and mutated sequences. Though with variations among individual sera, the trend that emerged from these analyses is that G37A, T38A, E39A, N40A and K41A affect the recognition of TSSAII[31-50] *N*-terminus whereas P42A, A43G, T44A, G45A, A47G, P48A and Q50 affect the recognition of TSSAII[31-50] *C*-terminus (S3 Fig). These results are consistent with our previous data (Fig 3) and, more importantly, allowed us to pinpoint certain invariant residues such as G37, K41, A43, G45 and P48 to either epitope A or B. The pattern obtained for low-impact P34A, P35A, and S36A replacements, on the other hand, was not clear (S3 Fig), though they were assigned to epitope A based on their relative position. Building on these findings, we redefined the sequences of epitopes A and B as 34-PPSGTENK-41 and 42-PATGEAPSQ-50, respectively.

Stratification of sera based upon their reactivity profile towards TSSAII (S1 Fig) allowed the formation of three groups: those displaying higher reactivity towards epitope A ($n = 18$), those showing similar recognition for both epitopes ($n = 11$), and those showing higher reactivity towards epitope B ($n = 3$) (Fig 6A). As expected, each group yielded different serum signatures upon mutation scan analyses (Fig 6C). Briefly, BR_P3 (epitope B-biased) revealed the sequence 40-N(K)PATGEAP(S)Q-50 (residues between parentheses were not included), which is highly coincident with our redefined epitope B (Fig 6C). BO_P3 and BR_P4 (unbiased), on the other hand, yielded 32-STPPSGTENKPATG(E)AP(S)Q-50 and 35-P(S)GTENKPATGEAPSQ-50 sequences, respectively, both of which contain residues from both epitopes, and AR_P6 (epitope A-biased) revealed a signature (34-PPSGTENKP-42) which is extremely similar to our redefined epitope A.

We also probed this mutational scanning array design with 33 sera from our panel that did not show primary reactivity towards TSSAII (Fig 2). As expected, the recognition of TSSAII[31-50] by these sera was negative or very low, and not affected by residue permutations (S4 Fig). However, and in contrast to the negative effect imposed by E39A permutation on sera showing primary reactivity to TSSAII, this substitution led to an increase in the signals recorded for this serum population (S4 Fig).

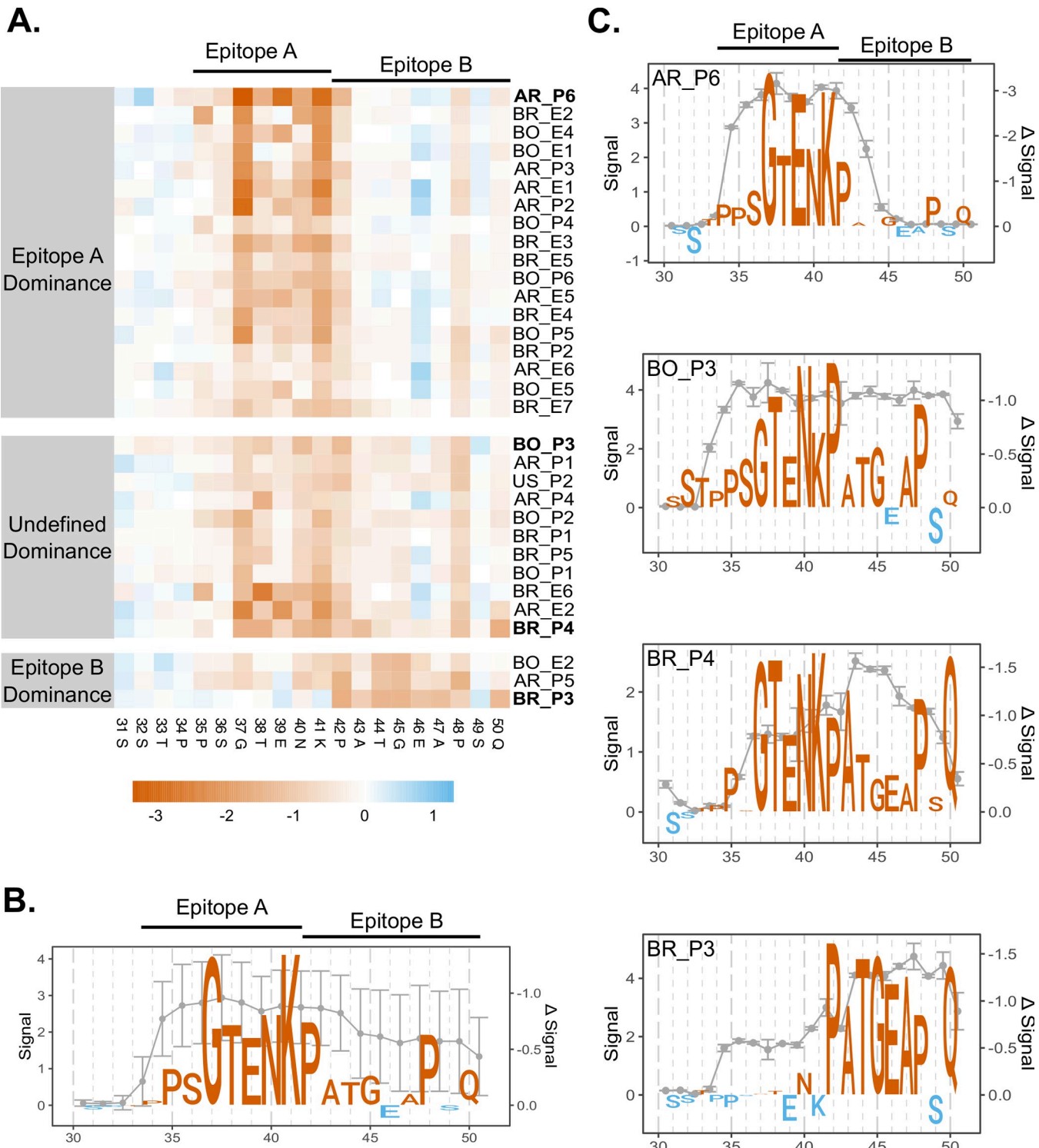

**Fig 6. *TSSAII mutational scanning.*** Peptide arrays comprising sets of fully overlapped 16mers, spanning residues 31 to 50 of TSSAII reference variant (TSSAII[31-50]) and bearing an Alanine substitution (or Glycine, in case of A43 and A47) at the indicated position were probed with 32 sera from individuals showing primary reactivity to TSSAII. **A)** Heatmap showing the overall impact of a specific replacement for each sample. Signal difference is represented by a color range for reactivity loss (orange) or gain (blue). Samples were divided into three groups based on their TSSAII reactivity profile (see S1 Fig). **B** and **C)** TSSAII reactivity profile and sequence logos representing the impact of each replacement for all analyzed sera (B) or for individual samples: AR_P6 (epitope A dominance), BO_P3 and BR_P4 (undefined dominance) and BR_P3 (epitope B dominance) (C). Peptide reactivity (left Y axis) is indicated by the mean ± SD values of biological (B) or technical replicates (C). The mean reactivity change caused by residue mutation (right Y axis) is indicated in orange (negative) or blue (positive). The proposed limits for epitopes A and B are indicated above the logos.

**Table 1. TcMARK TSSA reactivity in human populations.**

| Sample | GST | TSSAI[24-61] | TSSAIII[24-62] | TSSAIV[24-62] | TcMARK TSSA[30-50] | GST-Ag1 |
|---|---|---|---|---|---|---|
| CO_E2 | 0.03 ± 0.07[1] | 0.05 ± 0.07 | -0.05 ± 0.01 | **0.35 ± 0.17*** | 0.04 ± 0.02 | **1.21 ± 0.08*** |
| CO_E1 | 0.04 ± 0.03 | 0.04 ± 0.05 | -0.02 ± 0.01 | -0.01 ± 0.01 | 0.09 ± 0.04 | **0.41 ± 0.03*** |
| CO_P4 | 0.01 ± 0.05 | -0.01 ± 0.07 | -0.01 ± 0.06 | -0.03 ± 0.03 | 0.02 ± 0.01 | **0.61 ± 0.11*** |
| CO_P1 | -0.02 ± 0.01 | -0.03 ± 0.01 | -0.06 ± 0.03 | -0.02 ± 0.02 | 0.02 ± 0.04 | **0.31 ± 0.05*** |
| CO_E3 | 0.04 ± 0.01 | 0.08 ± 0.13 | 0.01 ± 0.06 | -0.02 ± 0.00 | **0.34 ± 0.06*** | **0.75 ± 0.00*** |
| CO_E4 | 0.19 ± 0.11 | 0.13 ± 0.07 | 0.09 ± 0.01 | 0.05 ± 0.00 | 0.14 ± 0.02 | **0.51 ± 0.03*** |
| CO_P3 | 0.03 ± 0.04 | 0.08 ± 0.07 | -0.01 ± 0.05 | -0.04 ± 0.01 | 0.03 ± 0.03 | **0.26 ± 0.04*** |
| CO_E6 | 0.08 ± 0.04 | 0.06 ± 0.00 | -0.06 ± 0.01 | -0.02 ± 0.04 | 0.05 ± 0.02 | **0.56 ± 0.10*** |
| CO_E7 | 0.01 ± 0.04 | 0.09 ± 0.01 | **0.76 ± 0.01*** | **1.26 ± 0.04*** | 0.07 ± 0.01 | **0.23 ± 0.10*** |
| CO_E5 | 0.06 ± 0.04 | 0.03 ± 0.02 | -0.06 ± 0.02 | -0.01 ± 0.04 | 0.04 ± 0.00 | **0.26 ± 0.04*** |
| CO_P2 | 0.04 ± 0.05 | 0.05 ± 0.00 | 0.00 ± 0.02 | -0.02 ± 0.02 | 0.04 ± 0.02 | 0.03 ± 0.01 |

[1] Values are expressed as mean ± SD values obtained from triplicates. One out of three experiments with similar results is shown. For each serum, reactivities were compared to GST values by 2-way ANOVA, using the Dunnett's multiple comparison.

* P Values < 0.0001.

## TcMARK TSSA and human serodiagnosis

To explore the serodiagnostic relevance of herein identified TcMARK TSSA, a GST-fusion protein bearing residues 30 to 50 of this molecule was probed by ELISA with serum samples from our panel of chronic Chagas disease patients from Argentina, Colombia, and the US. Signals were above the baseline determined for serum samples of noninfected individuals from the Colombian population. Reactivity of these samples against a panel of GST-fusion molecules is shown in Table 1. Evaluated antigens included, in addition to TcMARK TSSA[30-50], TSSAI[24-61], TSSAIII[24-62] and TSSAIV[24-62] as well as negative (GST) and positive (GST-Ag1) controls. ELISA results were very similar to those obtained by peptide microarrays, i.e., robust signals were only detected for the CO_E7 serum and the TSSAIII and TSSAIV molecules (compare Table 1 and Fig 2A). Most importantly, these assays revealed specific TcMARK TSSA recognition by the CO_E3 sample (Table 1).

## Discussion

*T. cruzi* is a very polymorphic taxon, comprising 6–7 evolutionary lineages showing variable eco-epidemiological traits. High intra-specific genetic variability may also have clinical implications, as distinct DTUs seem to be associated with differential susceptibility to drugs, disease features and infection outcomes [3,7–9,43], thus stressing the necessity of assigning the parasite strain type. Serological methods, relying on the profiling of host antibodies raised against polymorphic parasite antigens, emerge as a promising alternative to address this issue [14]. As extensively discussed, the *in vivo* performance of such methods is not curtailed by limitations intrinsic to *T. cruzi* genotyping schemes, including the very low and fluctuating parasitemia characterizing the chronic phase of the disease and the frequent occurrence of two or more parasite strains within the same infected host [23,25,43].

Epidemiological and clinical studies conducted so far have shown that, despite certain aspects that need to be improved, TSSA-based serotyping assays may provide robust, sensitive, low-cost and point-of-sampling diagnosis, with near DTU-level resolution [14]. We herein used peptide microarrays and samples from different settings to establish an exhaustive serological profiling of this polymorphic molecule. Our assays revealed variations in the seroprevalence and on the specificity of TSSA antibodies among Chagas disease populations. This is in

line with previous, TSSA-based epidemiologic studies, and strongly supports the differential distribution of strains/DTUs in domestic cycles across the Americas [2,3,5]. A low proportion (40%) of samples from Colombia, Mexico, and the US yielded TSSA reactivity, which was barely detectable and mostly directed to TSSAIV and/or TSSAI. On the contrary, 97.2% of samples from Argentina, Bolivia, and Brazil strongly recognized TSSAII. In addition, some of these samples displayed minor, secondary reactivity towards TSSAIII and/or TSSAIV variants. This might be attributed to the co-existence of antibodies to multiple TSSA variants in Argentina, Bolivia, and Brazil Chagas disease patients, suggestive of infection caused by hybrid strains, co-infection with different strains/DTUs or, alternatively, to antibody cross-recognition phenomena.

Consistent with previous data [27,28], fully overlapped 15mers unveiled a continuous immunoreactive region in TSSAII, which extended through peptides pS32 to pA53. Despite the complexity of the sample used, i.e., pools of sera containing multiple polyclonal antibodies with different specificities, evaluation of shorter kmers improved mapping resolution of this region, narrowing down immunoreactivity to peptides pP35 to pQ50 and allowing the identification of two antigenic peaks (named epitopes A and B) located at its boundaries. Solely the Brazilian pool, and most likely due to the inclusion of the BR_P2 serum, revealed a weak, third peak, which was proposed as epitope C. In some cases (e.g., the Argentina pool), epitopes A and B could not be fully resolved, which may be attributed to the close vicinity of the entailing sequences and/or to the presence of additional, intertwined antigenic determinants formed by residues from both sequences.

The existence of (at least) two discrete epitopes in TSSAII was further supported by residue permutation analyses that allowed the identification of diagnostically relevant or non-relevant substitutions among TSSA natural polymorphisms. Interestingly, relevant substitutions could be split into two functional categories: those affecting the reactivity of the *N*-terminal antigenic peak/epitope A (T38K, N40K) and those affecting the reactivity of the *C*-terminal antigenic peak/epitope B (P42A/T, T44A, E46G and A47T). Moreover, substitutions lying on the same epitope showed immunoreactive redundancy, i.e., yielded similar reactivity profiles when tested individually or in different combinations, strongly suggesting that all of them affect the recognition of a single antibody population. Mutational scanning studies provided further structural insights into TSSAII-antibody interaction and allowed us to refine the proposed sequences for epitope A (34-PPSGTENK-41) and B (42-PATGEAPSQ-50). These sequences fit snugly with previous mapping data that suggested the presence of at least two dominant and discrete linear B-cell epitopes in TSSAII, composed within residues 30 to 44 and 36 to 50, respectively [28].

In diagnostic terms, our findings indicate that epitope A is more relevant than epitope B. Indeed, reactivity of TSSAII-positive sera was either directed against both epitopes (with often lower signals for epitope B) or, less frequently, strongly biased towards epitope A (e.g., AR_P6). It should be however considered that epitope A is also more conserved across TSSA isoforms, and hence more prone to serological cross-recognition phenomena, which may undermine the performance of TSSA-based serotyping approaches. Therefore, novel designs of serodiagnostic reagents may need to separate epitopes A and B or even put them *in trans*, i.e., in different molecules, to better assess the specificity of anti-TSSA humoral responses and/or to avoid potential competition/displacement phenomena observed for different antibody populations recognizing adjacent epitopes [44]. Most interestingly, it may also be inferred from our data that suboptimal sensitivity of peptide TSSApep-II/V/VI (37-GTENKPAT-GEAPSQPG-52) included in the Chagas Sero K-SeT rapid diagnostic test [18,21] is due to the lack of residues P34, P35 and S36, intrinsic to epitope A.

Genome mining exercises revealed a larger-than-expected complexity of TSSA sequences, with two genomic *loci* in hybrid strains and tandems of several (up to 14) head-to-tail arranged

identical gene copies, usually intercalated by pseudogenes, in most *T. cruzi* genomes (S3 Table) [14]. We identified 9 novel TSSA variants, which were classified as TSSAII, TSSAIII, TSSAIV or TSSAI by clustering with reference sequences. In the Dm28c and Brazil A4 strains genomes [45,46], in addition, we detected 2 variants of the same TSSA isoform (TSSAI), which represents the first report of TSSA intra-strain heterogeneity in non-hybrid lineages. Puzzlingly, we could not detect the TSSAII-type allele bearing A44T substitution, claimed as diagnostic for hybrid DTUs [12].

Our findings reveal a strict correlation between the identified TSSA isoform(s) and the reported DTU; which strongly stresses the potential of this molecule as a genotypic marker [10]. Discordant cases included TSSAI sequences in one Y clone (TcII) and in the Bug2148 (TcV) strain, and TSSAII sequences in Ikiakarora strain (TcIII). It should be noted, however, that i) the rest of Y isolates sequenced so far yielded TSSAII sequences (see S3 and S4 Tables); and ii) comparative genomic analyses clearly indicated that Bug2148 clusters with TcI strains [46–48]. Interestingly, our comprehensive screening of genomic repositories also retrieved one TSSA sequence from the phylogenetically related bat parasite *T. cruzi marinkellei* [49], which raises doubts about the evolutionary origin and distribution of TSSA in the trypanosomatids' clade. Despite its evident genetic drift, this molecule shared quite similar structural features with *T. cruzi* TSSAs, such as protein length and overall amino acid composition. Moreover, *in silico* predictions indicated that TcMARK TSSA retained a *N*-terminal signal peptide and a canonical *C*-terminal GPI-anchoring signal, thereby suggesting that, as *T. cruzi* TSSAs, it is a surface localized and/or secreted protein [10,16]. Most importantly, specific serorecognition to TcMARK TSSA was recorded for one chronic Chagas disease patient from Colombia, and this recognition could not be attributed to serological cross-reactivity with *T. cruzi* TSSAs. Though preliminary, these findings warrant further investigations on the diagnostic impact of such atypical TSSA, especially given recent evidence of human infections with TcBat and/or other closely related bat trypanosomes [50,51].

Overall, our findings shed new light into TSSA antigenic make up and provide essential information for guiding and/or improving the design, resolution, specificity, and applicability of TSSA-based serotyping strategies. Dissection of the epitope mesh in TSSAII, in addition, might be further explored in other areas of Chagas disease epidemiology and diagnosis such as assessment of therapy efficacy and prognostic biomarker identification.

## Supporting information

**S1 Table. Information about the population study analyzed in this work.**
(DOCX)

**S2 Table. Features of the TSSA peptide microarrays.**
(DOCX)

**S3 Table. Features of TSSA sequences in Trypanosomatids.**
(DOCX)

**S4 Table. Sequences of previously undescribed TSSA genes.**
(DOCX)

**S5 Table. Sequences of GST-fusion TSSAs used in the *TSSA-ELISA*.**
(DOCX)

**S1 Fig. Mapping the antibody recognition of TSSA variants by Chagas disease sera.** Peptide arrays comprising sets of fully overlapped 16mers and encompassing the complete sequence of TSSAI, TSSAII, TSSAIII or TSSAIV reference variants were probed with 71 serum samples

from chronic Chagas disease patients of different geographic origin ('AR' = Argentina; 'BO' = Bolivia; 'BR' = Brazil; 'CO' = Colombia; 'MX' = Mexico; 'US' = United States). The mean reactivity of each residue in the context of individual sequences is indicated with a color scale. For each TSSA isoform, samples from each geographic origin were ordered according to their reactivity.
(TIFF)

**S2 Fig. Impact of mutation accumulation on the recognition of TSSAII$^{30-55}$ by Chagas disease sera.** Peptide arrays comprising sets of fully overlapped 15mers, encompassing residues 30 to 55 of TSSAII reference variant (TSSAII$^{30-55}$) and bearing selected substitution(s) at polymorphic positions were probed with a pool of sera of chronic Chagas disease patients from Bolivia. The mean reactivity of each residue in the context of wild-type (top row) and single- or multiple-substituted sequences is indicated with a color scale. For each sequence, the replaced residue(s) is/are indicated.
(TIFF)

**S3 Fig. Impact of Alanine substitutions on the recognition of TSSAII$^{31-50}$ by Chagas disease sera.** Peptide arrays comprising sets of fully overlapped 16mers, encompassing residues 31 to 50 of TSSAII reference variant (TSSAII$^{31-50}$) were probed with chronic Chagas disease sera from the indicated geographical origin. The net impact caused by selected Alanine substitutions (indicated in red) was calculated as the difference in reactivity for each residue between the original and mutated sequences. This difference in reactivity is indicated with a color scale. In the case of Alanine 43 (bottom left panel), it was mutated to G.
(TIFF)

**S4 Fig. Impact of E39A substitution on the recognition of TSSAII$^{31-50}$ by Chagas disease sera. A)** Heatmap showing the overall impact of a specific replacement on TSSAII$^{31-50}$ recognition by Chagas disease samples either showing (right panel, *n* = 32) or not showing (left panel, *n* = 33) primary reactivity to TSSAII (for further details see legend to Fig 6). In both panels, the line corresponding to the E39A replacement is highlighted. **B)** Impact of E39A substitution on the reactivity to TSSAII$^{31-50}$ of TSSA-negative serum samples (left panel) or from serum samples showing primary reactivity to TSSAI, TSSAIV or TSSAII. The reactivity of all peptides containing the E39A substitution is shown for each serum sample (in each case, the median ± SD are indicated by box and whiskers).
(TIFF)

**S1 File. Complete list of sequences and reactivities of TSSA peptides evaluated in Fig 2.**
(TSV)

**S2 File. Complete list of sequences and reactivities of TSSA peptides evaluated in Figs 3 and 5.**
(TSV)

**S3 File. Complete list of sequences and reactivities of TSSA peptides used for the mutational scanning analyses in Fig 6.**
(TSV)

## Acknowledgments

We thank Dr Luciano J. Melli (IIBio) for providing us with GST-Ag1. GR, LB and ADR hold fellowships from the National Research Council (CONICET, Argentina), and VB, JA, FA and CAB are career investigators from the same Institution.

## Author Contributions

**Conceptualization:** Guadalupe Romer, Leonel A. Bracco, Alejandro D. Ricci, Virginia Balouz, Carlos A. Buscaglia, Fernán Agüero.

**Data curation:** Guadalupe Romer, Leonel A. Bracco, Alejandro D. Ricci, Virginia Balouz, Carlos A. Buscaglia, Fernán Agüero.

**Formal analysis:** Guadalupe Romer, Leonel A. Bracco, Alejandro D. Ricci, Virginia Balouz, Carlos A. Buscaglia, Fernán Agüero.

**Funding acquisition:** Carlos A. Buscaglia, Fernán Agüero.

**Investigation:** Guadalupe Romer, Leonel A. Bracco, Alejandro D. Ricci, Virginia Balouz, Carlos A. Buscaglia, Fernán Agüero.

**Methodology:** Guadalupe Romer, Leonel A. Bracco, Alejandro D. Ricci, Virginia Balouz, Luisa Berná, Carlos Robello, Carlos A. Buscaglia, Fernán Agüero.

**Project administration:** Carlos A. Buscaglia.

**Resources:** Leonel A. Bracco, Luisa Berná, Juan C. Villar, Janine M. Ramsey, Melissa S. Nolan, Faustino Torrico, Norival Kesper, Jaime Altcheh, Carlos Robello, Carlos A. Buscaglia, Fernán Agüero.

**Supervision:** Carlos A. Buscaglia, Fernán Agüero.

**Validation:** Carlos A. Buscaglia, Fernán Agüero.

**Visualization:** Guadalupe Romer, Leonel A. Bracco, Carlos A. Buscaglia, Fernán Agüero.

**Writing – original draft:** Guadalupe Romer, Leonel A. Bracco, Alejandro D. Ricci, Virginia Balouz, Carlos A. Buscaglia, Fernán Agüero.

**Writing – review & editing:** Guadalupe Romer, Leonel A. Bracco, Alejandro D. Ricci, Virginia Balouz, Carlos A. Buscaglia, Fernán Agüero.

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
