## [Decision Letter · Decision Letter 0]

23 May 2023

Dear Dr Buscaglia,

Thank you very much for submitting your manuscript "Deep serological profiling of the Trypanosoma cruzi TSSA antigen reveals different epitopes and modes of recognition by Chagas disease patients" for consideration at PLOS Neglected Tropical Diseases. As with all papers reviewed by the journal, your manuscript was reviewed by members of the editorial board and by several independent reviewers. The reviewers appreciated the attention to an important topic. Based on the reviews, we are likely to accept this manuscript for publication, providing that you modify the manuscript according to the review recommendations. 

Please pay specific attention to the comments made by Reviewers 2 and 3.

Sincerely,

Charles L. Jaffe, Ph.D.

Section Editor

Charles Jaffe

Section Editor

Please pay specific attention to the comments made by Reviewers 2 and 3.

Reviewer's Responses to Questions

**Key Review Criteria Required for Acceptance?**

**Methods**

-Are the objectives of the study clearly articulated with a clear testable hypothesis stated?

-Is the study design appropriate to address the stated objectives?

-Is the population clearly described and appropriate for the hypothesis being tested?

-Is the sample size sufficient to ensure adequate power to address the hypothesis being tested?

-Were correct statistical analysis used to support conclusions?

-Are there concerns about ethical or regulatory requirements being met?

Reviewer #1: -Are the objectives of the study clearly articulated with a clear testable hypothesis stated? YES

-Is the study design appropriate to address the stated objectives? YES

-Is the population clearly described and appropriate for the hypothesis being tested? YES

-Is the sample size sufficient to ensure adequate power to address the hypothesis being tested? YES

-Were correct statistical analysis used to support conclusions? YES

-Are there concerns about ethical or regulatory requirements being met? NO

Reviewer #2: See Summary and General Comments

Reviewer #3: (No Response)

**Results**

-Does the analysis presented match the analysis plan?

-Are the results clearly and completely presented?

-Are the figures (Tables, Images) of sufficient quality for clarity?

Reviewer #1: -Does the analysis presented match the analysis plan? YES

-Are the results clearly and completely presented? YES

-Are the figures (Tables, Images) of sufficient quality for clarity? YES

Reviewer #2: See Summary and General Comments

Reviewer #3: (No Response)

**Conclusions**

-Are the conclusions supported by the data presented?

-Are the limitations of analysis clearly described?

-Do the authors discuss how these data can be helpful to advance our understanding of the topic under study?

-Is public health relevance addressed?

Reviewer #1: -Are the conclusions supported by the data presented? YES

-Are the limitations of analysis clearly described? YES

-Do the authors discuss how these data can be helpful to advance our understanding of the topic under study? YES

-Is public health relevance addressed? YES

Reviewer #2: See Summary and General Comments

Reviewer #3: (No Response)

**Editorial and Data Presentation Modifications?**

Reviewer #1: A few minor adjustments are recommended, as follows: 

Line 174-175. The authors should clarify in the legend whether the letters ‘A’, ‘C’ , ‘D’ etc, in Figure 1B are or are not the standard single-letter abbreviations for amino acids (as used in Figure 1A).

Line 227. The authors should state (n = 30), rather than simply ‘{30}’.

Line 282. Replace the word ‘above’ with ‘top’.

Line 515-516. Do the authors mean ‘Table 1’ here in the text (not 'Table 2')?

Table 1. In the column headings, the authors should add the superscript numbers to the TSSA peptides as done in the text, for example TSSAIV 24-62

Reviewer #2: See Summary and General Comments

Reviewer #3: (No Response)

**Summary and General Comments**

Reviewer #1: This is a detailed analysis of the Trypanosoma cruzi antigen TSSA, revealing new diversity, with relevance to geographical distribution and to diagnostic sensitivity. This rigorous research will contribute to understanding of the epidemiology of Chagas disease, and to improvement of regional diagnostic efficacy. 

The methods are complex, and results detailed. However, in the Discussion, some of the results are already described or are expected, for example, as with samples from TSSA1 and TSSAIV, for which it is well-known where these infections predominate. The two variants of TSSA1 are interesting and their epidemiology worthy of follow up, with wider comparative genomics.

There are no significant flaws, and the manuscript is worthy of publication.

Reviewer #2: The authors have a long history in exploring the serological response to the TSSA antigen for the identification of T. cruzi DTU infecting strain.

The manuscript describes a thoughtful work to characterize the linear B-cell epitope profiling of TSSA and reveals variations in the seroprevalence and on the specificity of TSSA antibodies among Chagas disease populations. The findings show a good correlation between the identified TSSA isoform(s) and the reported DTU.

Major points

1. Authors’ citations

- In the manuscript, extensive self-citation of co-authors is noted. Important references are not cited. I invite authors to revise the recent literature. For example, the review by Magalhães et al. 2022 (Lancet Microbe doi: 10.1016/S2666-5247(21)00265-2) on T. cruzi pathogen diversity, immunity, and the fate of infections should clearly be mentioned in the Introduction and Discussion.

- Introduction

Line 103. methods able to reliably assign the infecting strain

Authors should mention that molecular methods are available to genotype the infecting strain DTU and their limitations (see Zingales et al. 2012 MEEGID doi:10.1016/j.meegid.2011.12.009).

- Discussion

Line 530. “as distinct DTUs seem to be associated with differential susceptibility to drugs, disease features and infection outcomes [40]. Reference [40] does not illustrate all the information in the sentence. To this reference must be added [refs 6, 7] and 

Zingales 2018. Acta Tropica dx.doi.org/10.1016/j.actatropica.2017.09.017

Magalhães et al., 2022. Lancet Microbe doi: 10.1016/S2666-5247(21)00265-2.

- Reference citation.

The citation of references in the text must be standardized. Sometimes references are cited between square brackets, sometimes between parentheses, sometimes as a superscript.

2. The sera used are a fundamental element of the study. The authors must justify the choice and characterize the patients/sera.

- Justify the six countries whose sera were analyzed.

In the Abstract and in Line 139, it is mentioned “From different endemic settings” and "from different endemic areas". Regions of the USA cannot be considered “endemic”.

Please revise.

- The high number of human sera from the USA (n=18) draws attention.

Most likely the patients are migrants from Latin American countries. This aspect should be mentioned in the article.

- In M&M, Page 6. Line 149, the authors state “The complete information of the population study can be found in [28]”. This reference needs to be updated. Ricci et al. 2023 Nature Communications | (2023) 14:1850.

On the other hand, it is important that the present article reports for EACH COUNTRY the number of the analyzed sera as well as the gender distribution and the mean and standard deviation of age of the patients. It is known that the immune response to T. cruzi varies with these parameters. Please note that for the sera from Argentina the gender of the patients is not specified in Suppl Data 2 by Ricci et al. The sera were provided by Dr. Altcheh, co-author of the present manuscript.

- The authors claim that the patients were asymptomatic (indeterminate ChD form). What tests were performed for this diagnosis? Why is it important for the study that the patients are asymptomatic? Please, inform this in the article.

Reviewer #3: In the present study, Romer et al. performed a deep linear B cell epitope mapping on TSSA (trypomastigote small surface antigen), as adhesin the elicit a strong antibody response during T. cruzi infections. Sera from Chagas disease patients from distinct endemic settings in the Americas, where different parasite DTUs circulate, were used to comprehensively evaluate the reactivity of peptides derived from TSSA sequences from different parasite isolates available in public databases. TSSA isoforms displayed different seroprevalence among Chagas disease patients correlating with differential distribution of parasite DTUs in the Americas. Ala-scan approach allow the identification of critical residues involved in the antibody binding. New TSSA variants and epitopes were identified, including a T. cruzi marinkellei sequence that reacted with serum from a Chagas disease patient from Colombia. This paper represents an important contribution for the study of TSSA linear B-cell epitopes and deserves publication after some aspects are clarified.

Lines 146-149: 

Sample naming: What is the letter “E” in the suffix?

Please, provide more information regarding the microarray design, including the complete list of peptides that were evaluated in this study and the raw fluorescence values for each assayed peptide in each experiment.

Provide the public accession numbers for all the TSSA reference sequences (Figure 1). 

Legend figure 2: Please state what is the panel “isoforms recognized”

Specify the genome versions as well as the public accession numbers included in the phylogenetic analysis. For genomes not publicly available, please provide the accession number of each gene/protein sequence used in this study or provide all sequences as supplementary material data.

Please provide the nucleotide sequence of the constructs of the (GST)-fusion proteins as supplemental material.

Please include in the Table S2 the database from which each sequence was retrieved.

Please describe the GST-Ag1 (control) used in the Table 1.

Typo: 

Line 64: Trypanosoma cruzi: italic

PLOS authors have the option to publish the peer review history of their article (what does this mean?). If published, this will include your full peer review and any attached files.

Reviewer #1: No

Reviewer #2: No

Reviewer #3: No

Figure Files:

Data Requirements:

Reproducibility:

References

---

## [Decision Letter · Decision Letter 1]

10 Jul 2023

Dear Dr Buscaglia,

Thank you very much for submitting your manuscript "Deep serological profiling of the Trypanosoma cruzi TSSA antigen reveals different epitopes and modes of recognition by Chagas disease patients" for consideration at PLOS Neglected Tropical Diseases. As with all papers reviewed by the journal, your manuscript was reviewed by members of the editorial board and by several independent reviewers. The reviewers appreciated the attention to an important topic. Based on the reviews, we are likely to accept this manuscript for publication, providing that you modify the manuscript according to the review recommendations. 

Sincerely,

Charles L. Jaffe, Ph.D.

Section Editor

Charles Jaffe

Section Editor

Reviewer's Responses to Questions

**Key Review Criteria Required for Acceptance?**

**Methods**

-Are the objectives of the study clearly articulated with a clear testable hypothesis stated?

-Is the study design appropriate to address the stated objectives?

-Is the population clearly described and appropriate for the hypothesis being tested?

-Is the sample size sufficient to ensure adequate power to address the hypothesis being tested?

-Were correct statistical analysis used to support conclusions?

-Are there concerns about ethical or regulatory requirements being met?

Reviewer #1: Revised amendments OK

Reviewer #2: The revised manuscript meets the criteria.

Reviewer #3: (No Response)

**Results**

-Does the analysis presented match the analysis plan?

-Are the results clearly and completely presented?

-Are the figures (Tables, Images) of sufficient quality for clarity?

Reviewer #1: Revised amendments OK

Reviewer #2: The revised manuscript meets the criteria.

Reviewer #3: (No Response)

**Conclusions**

-Are the conclusions supported by the data presented?

-Are the limitations of analysis clearly described?

-Do the authors discuss how these data can be helpful to advance our understanding of the topic under study?

-Is public health relevance addressed?

Reviewer #1: Revised amendments OK

Reviewer #2: The revised manuscript meets the criteria.

Reviewer #3: (No Response)

**Editorial and Data Presentation Modifications?**

Reviewer #1: Revised amendments OK

Reviewer #2: No modifications. Accept revised manuscript.

Reviewer #3: (No Response)

**Summary and General Comments**

Reviewer #1: Revised amendments OK

Reviewer #2: The author responded to points raised by the reviewer and, accordingly, modifications were made to the manuscript.

Reviewer #3: Two points are still pending:

1- The accession numbers of the Figure 1 sequences are still missing (they are not in the legend of Figure 1). 

2- Please provide the complete list of TSSA peptide sequences and their raw fluorescence values evaluated in this study as a supplementary file that will be hosted in the PNTD website (and not in the github author’s account).

PLOS authors have the option to publish the peer review history of their article (what does this mean?). If published, this will include your full peer review and any attached files.

Reviewer #1: No

Reviewer #2: No

Reviewer #3: No

Figure Files:

Data Requirements:

Reproducibility:

References

---

## [Editor Report · Decision Letter 2]

18 Jul 2023

Dear Dr Buscaglia,

We are pleased to inform you that your manuscript 'Deep serological profiling of the Trypanosoma cruzi TSSA antigen reveals different epitopes and modes of recognition by Chagas disease patients' has been provisionally accepted for publication in PLOS Neglected Tropical Diseases.

Best regards,

Charles L. Jaffe, Ph.D.

Section Editor

Charles Jaffe

Section Editor

---

## [Editor Report · Acceptance letter]

3 Aug 2023

Dear Dr Buscaglia,

We are delighted to inform you that your manuscript, "Deep serological profiling of the *Trypanosoma cruzi* TSSA antigen reveals different epitopes and modes of recognition by Chagas disease patients," has been formally accepted for publication in PLOS Neglected Tropical Diseases.

Best regards,

Shaden Kamhawi

co-Editor-in-Chief

Paul Brindley

co-Editor-in-Chief
